# Supernumerary Marker Chromosome Identified in Asian Elephant (*Elephas maximus*)

**DOI:** 10.3390/ani13040701

**Published:** 2023-02-17

**Authors:** Halina Cernohorska, Svatava Kubickova, Petra Musilova, Miluse Vozdova, Roman Vodicka, Jiri Rubes

**Affiliations:** 1Department of Genetics and Reproductive Biotechnologies, Veterinary Research Institute, 62100 Brno, Czech Republic; 2Zoo Praha, 17100 Praha, Czech Republic

**Keywords:** small supernumerary marker chromosome, sSMC, laser microdissection, FISH, karyotype, heterochromatin, NOR, Asian elephant, savanna elephant

## Abstract

**Simple Summary:**

Supernumerary marker chromosomes, as they are known in the human population, are usually small chromosomes that differ morphologically and structurally from the standard ones and in many cases are formed by genetically inert heterochromatin. Similar features were observed for a supernumerary chromosome discovered in two Asian elephants, a mother and her male offspring. In this study, we present its detailed analysis using several molecular cytogenetic techniques including laser microdissection and fluorescence in situ hybridization that allowed identification of this marker chromosome. Based on our findings, we propose the most possible mechanism for the origin of the marker studied. We extended our investigation and showed that the distribution of nucleolar organizer regions on the chromosomes of Asian and savanna elephants may be related to the distribution of heterochromatin. Supernumerary chromosomes or, in other words, additional or extra chromosomes added to typical human or animal karyotypes, have recently gained the attention of scientists as model systems for the study of chromosome evolution, which may include the chromosome marker described here.

**Abstract:**

We identified a small, supernumerary marker chromosome (sSMC) in two phenotypically normal Asian elephants (*Elephas maximus*): a female (2n = 57,XX,+mar) and her male offspring (2n = 57,XY,+mar). sSMCs are defined as structurally abnormal chromosomes that cannot be identified by conventional banding analysis since they are usually small and often lack distinct banding patterns. Although current molecular techniques can reveal their origin, the mechanism of their formation is not yet fully understood. We determined the origin of the marker using a suite of conventional and molecular cytogenetic approaches that included (a) G- and C-banding, (b) AgNOR staining, (c) preparation of a DNA clone using laser microdissection of the marker chromosome, (d) FISH with commercially available human painting and telomeric probes, and (e) FISH with centromeric DNA derived from the centromeric regions of a marker-free Asian elephant. Moreover, we present new information on the location and number of NORs in Asian and savanna elephants. We show that the metacentric marker was composed of heterochromatin with NORs at the terminal ends, originating most likely from the heterochromatic region of chromosome 27. In this context, we discuss the possible mechanism of marker formation. We also discuss the similarities between sSMCs and B chromosomes and whether the marker chromosome presented here could evolve into a B chromosome in the future.

## 1. Introduction

The occurrence of a small supernumerary marker chromosome (sSMC) in the human karyotype is relatively rare and its identification is always difficult using standard cytogenetic methods [1]. This is because sSMCs represent a heterogeneous group of derivative chromosomes in terms of their chromosomal origin and shape as well as their clinical consequences [2]. It is estimated that in the human population, approximately 0.044% of newborn children are sSMC carriers [3]. About half of sSMCs are represented by heterochromatic markers that are usually harmless to their carriers. Most of them are derived from short arms and pericentric regions of acrocentric chromosomes, in which the most implicated acrocentrics are HSA15 (https://cs-tl.de/DB/CA/sSMC/0-Start.html; (accessed on 14 January 2023) [4]. The risk of an abnormal phenotype for the carrier depends on factors such as the size of the marker, genetic content, and level of mosaicism [5]. Approximately one third of the published cases correlate with specific clinical signs and symptoms, such as Emanuel, Pallister-Killian, Turner, or cat eye syndromes, while two-thirds of cases have not been associated with clinical abnormalities [5,6]. Because of the wide variety of marker chromosomes in the human population, it remains difficult to correlate a particular sSMC with a particular phenotype, especially in de novo cases [7]. Recently, it has been suggested that approximately 77% are de novo mutations, while 23% are inherited either maternally (16%) or paternally (7%) [8]. In most of the familial cases, there is no discernible increased risk of offspring abnormalities if one parent has the same marker and their phenotype is normal [9]. Familial sSMCs are preferentially maternally transmitted [10,11,12,13,14,15], suggesting either reduced fertility in male carriers or that the marker is excluded in spermatogenesis [14].

Even though numerous studies have been published on human cytogenetics, the presence of sSMCs in animals has not been reported to our knowledge. We identified a small metacentric marker chromosome in two phenotypically normal Asian elephants, which is undoubtedly a karyotypic novelty within elephants. Living elephantids (*Elephantidae* family) include three species: (a) two species of the genus *Loxodonta*, the savanna elephant (*Loxodonta africana*) and forest elephant (*Loxodonta cyclotis*), which are restricted to Africa, and (b) one species of the genus *Elephas*, the Asian or Indian elephant (*Elephas maximus*), which is endemic to Asia [16]. The latter species is of considerable economic significance in many Asian countries. Chromosomal data based on G- and C- banding and comparative FISH available for *E. maximus* and *L. africana* show a high level of chromosome band homology [17,18,19]. Their karyotypes possess 56 chromosomes and differ only in the amount and distribution of C-band positive heterochromatin [19].

In the present study, we report the outcome of a detailed molecular cytogenetic dissection of the marker chromosome and its identification. Moreover, we provide new information about the location and number of NORs and distribution of heterochromatin in Asian and savanna elephants. We hypothesize that the sSMC identified in this study might have some features that could contribute to its future development into a B chromosome.

## 2. Materials and Methods

### 2.1. The Asian Elephant Family

The female Asian elephant originating from the Pinnawala Elephant Orphanage, Sri Lanka Island, was imported to Prague Zoo (Prague, Czech Republic). In the Prague Zoo, she gave birth to two calves, a male and a female, who were sired by two different males. The pedigree chart is presented in Figure 1. All members of the elephant family were cytogenetically examined except the male who sired the male offspring because he is currently kept in a Zoo in Switzerland and his karyotype is not available.

### 2.2. Samples and Banding Techniques

Peripheral blood samples were collected from four Asian elephants (*E. maximus*, EMA) held in the Prague Zoo: two females and two males. A blood sample was also taken from the female savanna elephant (*L. africana*, LAF) held in the Dvur Kralove Zoo. Blood samples were collected by zoo veterinarians for the purpose of preventive examinations or other medical procedures and an aliquot of the blood was used for cytogenetic studies. Metaphase spreads were prepared using culture protocols described by Cernohorska et al. [20]. Conventional protocols for G- and C-banding and AgNOR staining followed Seabright [21], Sumner [22], and Goodpasture and Bloom [23], respectively. The G-banded karyotype of the Asian elephant was arranged according to Yang et al. [19].

### 2.3. DNA Probes and Fluorescence In Situ Hybridization (FISH)

#### 2.3.1. Preparation of the EMAM1 Clone

We used the PALM Microlaser system (Carl Zeiss MicroImaging GmbH, Munich, Germany) to collect 20 copies of the marker chromosome. DNA of the collected chromosomes was amplified by degenerate oligonucleotide primed polymerase chain reaction (DOP-PCR), labeled during the secondary PCR with Orange-dUTP (Abbott, IL, USA) as described by Kubickova et al. [24] and checked by FISH. Amplification products derived from the marker were cloned into a pDrive vector (Qiagen, Hilden, Germany). The clones were screened by DOT-BLOT hybridization [25], fluorescently labeled by Orange-dUTP, and checked for specificity by FISH. Plasmid DNA of the selected clone was subsequently isolated and sequenced by Sanger sequencing. The clone comprised repetitive DNA but was not long enough to represent a basic repeat unit (BRU). Therefore, primers amplifying the 5′- and 3′- flanking regions were designed and inverse PCR was performed on the genomic DNA [20]. The amplification products representing the BRU obtained by PCR were cloned and the plasmid DNA was isolated, fluorescently labeled by Orange-dUTP, and used in the FISH analysis. The BRU clone was named EMAM1 clone, sequenced, and deposited in GenBank under accession number OP918028.

#### 2.3.2. Preparation of the LAFM1 Clone

Primers selected for inverse PCR in *E. maximus* were used on *L. africana* genomic DNA to obtain the BRU. The amplification products were cloned and the plasmid DNA was isolated, labeled by Orange-dUTP, and checked for specificity by FISH (see the procedure described above). One clone was chosen based on fluorescence intensity and sequenced. The BRU was named LAFM1 clone and deposited in GenBank under accession number OP918029. The sequences of both clones were compared using BLAST2 software and screened for interspersed repeats using RepeatMasker (https://www.repeatmasker.org/cgi-bin/WEBRepeatMasker; (accessed on 20 May 2019).

#### 2.3.3. Preparation of the Centromeric Probe

For generation of the centromeric probe, the DNA template was taken from the centromeric regions of the selected marker-free Asian elephant chromosomes by laser microdissection. The pooled DNA was amplified by DOP-PCR, labeled during the secondary PCR with Orange-d UTP, and checked by FISH [24].

#### 2.3.4. Telomere-Specific Probe

A commercially available Telomere PNA/FITC probe (DAKO A/S, Glostrup, Denmark) was used for FISH following the manufacturer’s recommendations.

#### 2.3.5. FISH

FISH procedures for chromosome painting and specific probes followed previously described protocols [20,26]. Hybridization signals were examined using Zeiss Axio imager.Z2 fluorescence microscope with appropriate fluorescent filters; images were captured by a CoolCube CCD camera (MetaSystems, Altlussheim, Germany) and analyzed by ISIS (MetaSystems).

#### 2.3.6. Identification of the NOR-Bearing Chromosomes in *E. maximus* and *L. africana*

NOR-bearing chromosomes were identified by FISH using human whole chromosome painting probes. The chromosome correspondence between human and elephants (Asian, EMA and African, LAF) was inferred from the comparative chromosome map established by Yang et al. [19]. A subset of Green- or Orange-labeled commercially available human chromosome-specific probes (MetaSystems, Altlussheim, Germany) were applied to both *E. maximus* and *L. africana* chromosomes following the hybridization protocol described by Yang et al. [19]. After hybridization, digital images were captured and the slides were subsequently treated with AgNOR staining [23]. The obtained images were compared to the FISH results to identify the NOR bearing chromosomes.

## 3. Results

Chromosome G-banding revealed a supernumerary chromosome in the elephant mother (2n = 57,XX,+mar) (Figure 2) and her son (2n = 57,XY,+mar). The marker was present in all metaphases examined in both animals (we examined 100 cells per animal). The diploid chromosome number of both the daughter and her father was 2n = 56, which was in accordance with the normal Asian elephant karyotype [17,27,28]. The marker chromosome was identified as small, metacentric (Figure 2 and Figure 3), and C-band positive (Figure 3) with NOR sites at the terminal ends (Figure 4a). Hybridization with the telomeric probe showed (a) strong signals at the terminal ends of all *E. maximus* chromosomes including the marker, (b) weak signals at the centromeric region of biarmed chromosomes, and (c) a strong signal in the central constriction of the biarmed marker (Figure 5a). Hybridization with the centromeric probe showed signals at the centromeric regions of most *E*. *maximus* chromosomes. The centromeric regions of the biarmed autosomes including the marker were not painted (data not shown).

### 3.1. Identification of the Marker Origin

In order to identify the origin of the marker chromosome, we applied several conventional and molecular cytogenetic methods. FISH with the EMAM1 probe prepared by microdissection of the marker revealed strong signals at the marker and p-arms of two small autosomal pairs, one metacentric and one submetacentric, which appeared almost entirely heterochromatic upon C-banding. Weaker signals were observed at the centromeric regions of several other autosomes (Figure 3 and Figure 5b). We identified the two autosomal pairs with heterochromatic p-arms using human (*Homo sapiens*, HSA) painting probes. The q-arm of the small metacentric pair was painted by the HSA2 probe and corresponded to the EMA27 chromosome on the comparative map. The q-arm of the small submetacentric pair was painted by the HSA13 probe corresponding to EMA16 (Figure 6). The subsequent silver-staining revealed that the terminal ends of the marker and heterochromatic p-arms of EMA27 and EMA16 possessed NORs (Figure 4a). Both FISH and NOR staining results indicated that the marker may have originated either from the EMA27 or EMA16 chromosomes. Based on the amount of heterochromatin included into the marker, it seems reasonable to suggest that the marker originated from p-arms of EMA27 rather than EMA16 (Figure 3 and Figure 5b). On closer inspection, variation in the amount of heterochromatin in EMA27 chromosomes was found in all of the Asian elephants examined. The metacentric shape of the marker suggested the isochromosome nature of the sSMC.

### 3.2. Identification of the NOR-Bearing Chromosomes Using Human Painting Probes and AgNOR Staining

#### 3.2.1. *E. maximus*

The NOR positions determined in both marker carrier and marker-free Asian elephants were on four autosomal pairs and their number ranged from 5 to 8 in the metaphases examined (Figure 4a). The NORs, all terminal, were located on the heterochromatic p-arms of EMA16 and EMA27 (see above) and q-arms of two acrocentric autosomal pairs corresponding to HSA17 (EMA11) and HSA18 (EMA13).

#### 3.2.2. *L. africana*

The NOR sites detected in the savanna elephant were located on nine terminal regions of eight autosomal pairs and their number varied from 11–16 in the metaphases examined (Figure 4b). The NORs, all terminal, were identified on: (a) the minute heterochromatic p-arms of six autosomal pairs corresponding to HSA1 (LAF2), HSA5 (LAF3), HSA4 (LAF5), HSA2 (LAF6), HSA13 (LAF16), and HSA2 (LAF27); (b) the q-arms of the autosomal pair corresponding to HSA17 (LAF11), and (c) both p- and q-arms of the autosomal pair corresponding to HSA18 (LAF13).

### 3.3. Comparison of the EMAM1 and LAFM1 Clones

The sequence homology of the BRU clones obtained from the Asian (EMAM1, 2619 kb in length) and savanna (LAFM1, 2629 kb in length) elephants was high (94%) and both clones lacked interspersed repeats (i.e., SINE, LINE or LTR elements), as revealed by RepeatMasker. We designated both sequences as satellite DNA based of the fact that the organization of repeat units in head-to-tail (or tandem) fashion permitted inverse PCR amplification [29]. The hybridization results with the EMAM1 probe to the *E. maximus* (Figure 5b) chromosomes are mentioned above. Hybridization with the LAFM1 probe to the *L. africana* chromosomes resulted in positive signals in the short p-arms of about half of the autosomes, whereas other chromosomes were unlabeled. (Figure 5c).

## 4. Discussion

Detection of an sSMC in phenotypically normal individuals is almost always an unexpected result in cytogenetic analysis and several molecular cytogenetic techniques are usually needed for their characterization. Using laser microdissection of the marker with subsequent reverse FISH that permitted the identification of the marker origin, we provide the first finding of an sSMC in wild mammals, to our knowledge. In the elephant family, the male offspring inherited the supernumerary chromosome from his mother while her daughter did not. Our observation fit the recently outlined fact that in humans, familial sSMCs are predominantly transmitted through the maternal line and familial marker chromosomes are usually harmless to their carriers [6]. Our marker is metacentric and only one central constriction was apparent. Since the supernumerary chromosome is mitotically stable, it presumably contains a functional centromere, even though the centromeric sequences were not detected with the centromeric probe, as with other biarmed chromosomes. The mirror-image shape of the marker indicates that the marker could have arisen in the same manner as isochromosomes [30]. The most possible explanation for the formation of our marker is that it might have originated in any ancestor during meiosis with an initial break in the (peri)centromeric region of EMA27 followed by horizontal separation of the p- and q-arms giving rise to isochromosomes i(27p) and i(27q). The separated heterochromatic p-arms, i(27p), later formed the marker chromosome (Figure 7). The strongest support for this explanation comes from the FISH results using a telomeric probe, which showed stronger fluorescence in the centromeric region of the marker in comparison to EMA27 (terminal telomeres are not considered here). As a consequence of the joining of the sister heterochromatic arms, the telomeric sequences located in the pericentromeric region appeared very close to one another, amplifying the signal in the center (Figure 5a). The nondisjunction of EMA27p during the first meiotic division subsequently resulted in maturation of an abnormal gamete, leading to the abnormal zygote with the heterochromatic sSMC after fertilization (Figure 7).

### 4.1. NORs

The marker chromosome detected in our study was heterochromatic with NORs at the terminal ends. In order to identify the origin of the marker, we determined chromosomes bearing both heterochromatin and NORs using a combination of AgNOR staining and FISH with human painting probes. Previous studies have used AgNOR staining on chromosomes of only *E. maximus* [31,32] without the identification of NOR-bearing chromosomes. Here, we present this information for *E. maximus* and also include the data for *L. africana*. Although the karyotypes of both species are largely conserved [19], the location and number of NORs show differences. Four NOR-bearing sites found in *E. maximus* (11q, 13q, 16p, and 27p) were shared by *L. africana*. Five other NOR sites identified in *L. africana* (2p, 3p, 5p, 6p, and 13p) were not found in *E. maximus*. In both species, NOR sites seem to be preferentially associated with heterochromatin, suggesting that the expansion of heterochromatic regions in *L. africana* might be related to the expansion of NORs and vice versa; the reduction of heterochromatic regions in *E. maximus* might be related to the reduction of NOR sites in the species.

### 4.2. Can We Consider the Marker Chromosome Identified in Two Asian Elephants as a Kind of Proto-B Chromosome?

There is some recent discussion in the literature about possible similarities between supernumerary marker chromosomes described in human and B chromosomes, which are enigmatic elements in eukaryotic karyotypes [33,34]. Both represent additional material to the main karyotype and may consist of heterochromatin and/or euchromatin. They are generally small in size and often lack specific phenotypic effects on the organisms that carry them. They are predominantly maternally transmitted and may be prone to mitotic instability [1,3,6,7,34,35]. It is estimated that B chromosomes occur in approximately 15% of eukaryotic species, the vast majority of which have been discovered in plants [36]. Other species in which B chromosomes have evolved include fungi, insects, helminth parasites, crustaceans, fish, amphibians, reptiles, birds, and mammals [35]. For the last group, up to 85 species carrying B chromosomes have been listed [37]. We reviewed the available literature in an attempt to determine whether the marker chromosome identified in our study in Asian elephants could evolve into a B chromosome in the future. It is believed that the best candidates for future B chromosomes in humans are (i) genetically inert (heterochromatic) supernumerary chromosomes that might manage to drive in either sex [34], (ii) sSMCs on which only their own DNA hybridizes [38], and (iii) acrocentric-derived inverted duplication sSMC with normal phenotypes [1,3,4]. Due to its heterochromatic nature, phenotypic inertness, and ability to be transmitted from parent to offspring, the marker chromosome identified in the two elephants in the current study can be included among the proposed candidates. In the recent review by Vujošević et al. ([37] and references in the article), the authors summarized that a typical B chromosome in mammals is seen as supernumerary, heterochromatic, smaller, and morphologically different from chromosomes of the standard set and does not evoke visible phenotypic effects. The size of the most common B chromosomes in mammals corresponds to the size of the smallest chromosome in the genome (occurring in 52 species, 65%), with metacentric and submetacentric shapes being more common than acrocentric shapes in this group. B chromosomes usually contain various repetitive sequences originally derived from autosomes, among which ribosomal and telomeric sequences have also been identified. Based on its morphological and molecular structure (i.e., small metacentric, heterochromatic chromosome with telomeric and rDNA sequences) and regardless of possible fertility disorders of their carriers, it seems that the marker chromosome identified in the current study has the potential to evolve into a B chromosome in the future. Additional rDNA sequences and/or other repeats or sequences on the marker may give some selective advantage to the carrier and thus may spread in a population of Asian elephants. In the future, it would be useful to collect DNA from the marker chromosome by microdissection and use it as a template for sequencing [39], as knowing the DNA content in the marker could help us determine how these chromosomes are formed.

## 5. Conclusions

We describe here the finding of a small supernumerary marker chromosome in a female elephant and her male offspring. Both animals were phenotypically normal, as is the case for most human carriers of markers containing heterochromatin. The fertility of the female did not appear to be affected by marker carriage, as she gave birth to two healthy offspring. Currently, we do not have any information about the fertility of the male offspring because he is not yet sexually mature. However, before being included in the captive breeding population, we recommend that his examination should entail microscopic and cytogenetic evaluation of the semen sample. Supernumerary marker chromosomes, especially B chromosomes, have recently gained the attention of scientists as model systems for the study of chromosome evolution, so it would be interesting to follow the fate of both elephants and their offspring in the future given that they carry a unique supernumerary chromosome that is not detrimental to their health and fitness.

## Figures and Tables

**Figure 1 animals-13-00701-f001:**
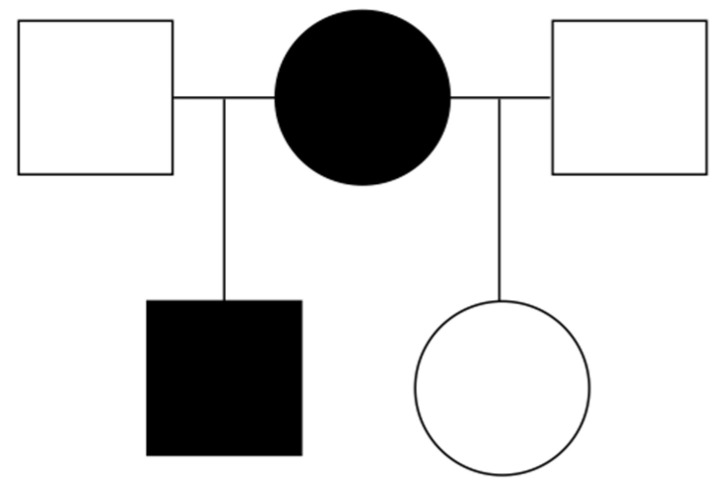
Pedigree chart for the Asian elephant family. Circles represent females and squares represent male individuals. sSMC carriers are marked in black.

**Figure 2 animals-13-00701-f002:**
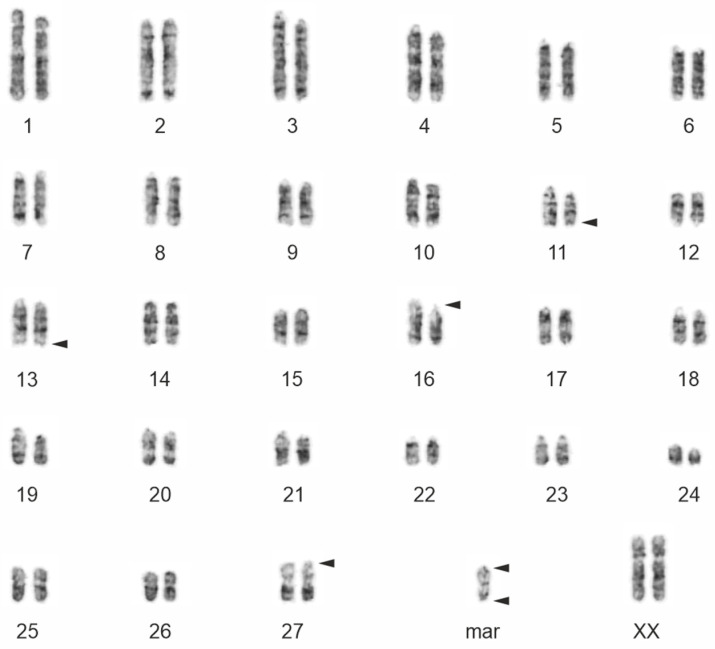
G-banded karyotype of *Elephas maximus* (2n = 57,XX,+mar). The chromosomes were arranged according to Yang et al. [19]. The arrowheads show the NOR positions.

**Figure 3 animals-13-00701-f003:**
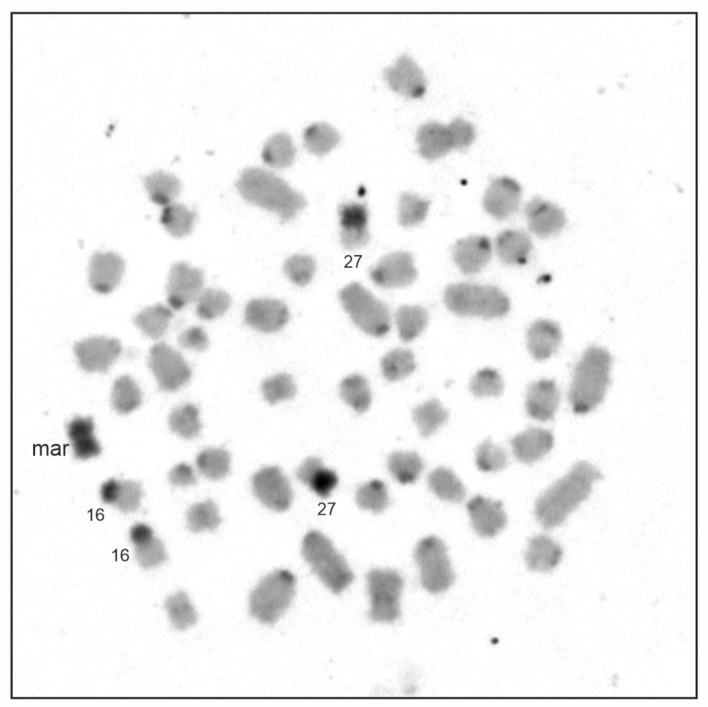
C-banded chromosomes of *E. maximus* (2n = 57,XX,+mar).

**Figure 4 animals-13-00701-f004:**
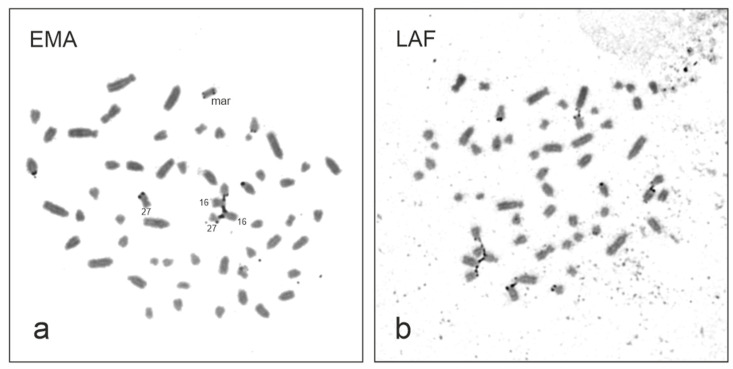
NOR positions in (**a**) *E. maximus* (2n = 57,XY,+mar) and (**b**) *L. africana* (2n = 56,XX).

**Figure 5 animals-13-00701-f005:**
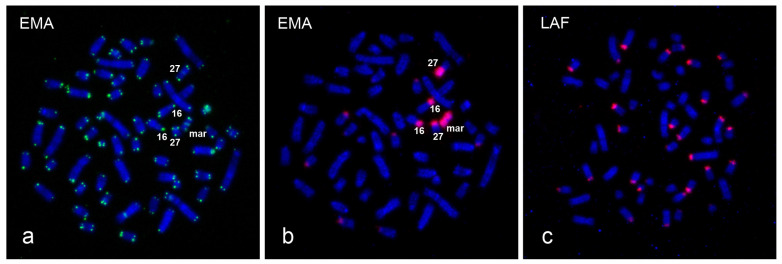
(**a**) FISH of a telomeric probe (green) to the *E. maximus*, EMA (2n = 57,XY,+mar). (**b**) The same metaphase spread hybridized with the EMAM1 probe (red). (**c**) FISH of the LAFM1 probe (red) to the *L*. *africana*, LAF chromosomes (2n = 56,XX). The chromosomes are counterstained with DAPI (blue).

**Figure 6 animals-13-00701-f006:**
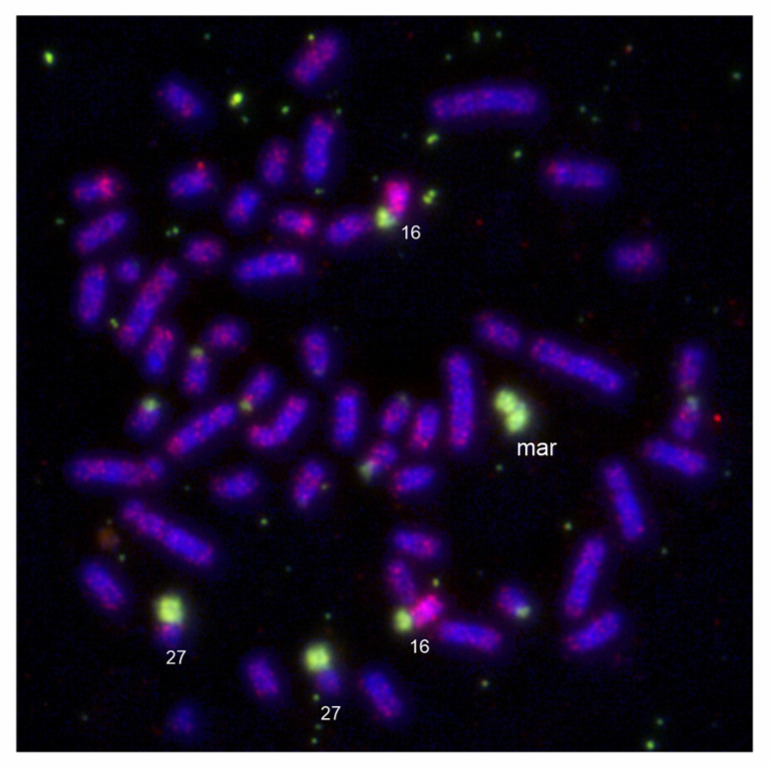
Co-hybridization of the EMAM1 (green) and HSA13 probes (red) to metaphase chromosomes of *E. maximus* (2n = 57,XX,+mar). The HSA13 probe shows signals on the EMA16 chromosome. The chromosomes are counterstained with DAPI (blue).

**Figure 7 animals-13-00701-f007:**
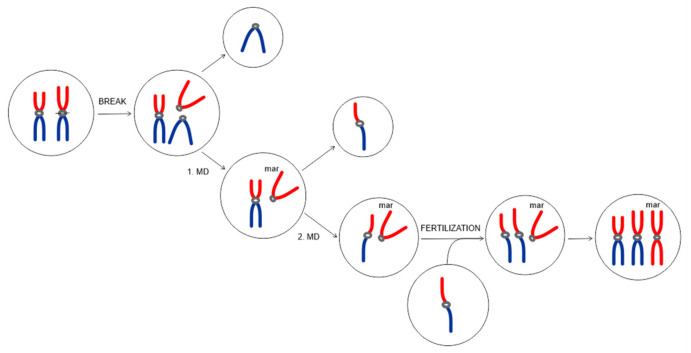
Schematic reconstruction showing the most possible marker formation during meiosis. The initial break in the (peri)centromeric region of one of the EMA27 homologs led to the horizontal separation of the p- and q-arms producing two isochromosomes, i(27p) and i(27q). One product of the misdivision was the sSMC formed by heterochromatic p-arms, i(27p). The other product was formed by the q-arms of the chromosome, i(27q). During the first meiotic division (MD), the marker along with the normal EMA27 chromosome segregated into one daughter cell. During the second MD, the sister chromatids of the normal EMA27 were released and segregated from one another. The marker chromosome along with one of the normal EMA27 chromatids segregated into the daughter cell, giving rise to an abnormal gamete, and leading to the abnormal zygote with the heterochromatic sSMC after fertilization.

## Data Availability

All data is contained within the manuscript. The FISH images are available from the authors upon request.

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
