# Peer review of "Supernumerary Marker Chromosome Identified in Asian Elephant (Elephas maximus)"

_animals, 2023, doi:10.3390/ani13040701_

Round 1

Author Response

Reviewer 1

Thank you for reviewing the manuscript. We addressed all your comments and made changes to the manuscript.

R: Lines 83 and 34: Change “NOR” with “Ag-NOR”

A: We have changed it in the text.

R: Line 139: How many metaphases were examined in both animals?

A: We have included this information to the text: “(we examined 100 cells per animal)”.

R: Please report in italics all Latin names of species

A: We have done it. The text originally contained all Latin names in italics, but they were changed during the submission of the article.

R: References in the text should follow the journal recommendations and list of references should be arranged according of their appearance in the text.

A: We have arranged the references according to the journal recommendations.

Reviewer 2 Report

In a well-founded and excellently written paper the authors report finding of supernumerary marker chromosome (sSMC) in two phenotypically normal Asian elephants, Elephas maximus. It is the first finding of supernumerary marker chromosomes in animal species other than humans. Applying both conventional and appropriate molecular techniques to analyze structure and origin of these sSMCs.  They found that small metacentric marker is composed of heterochromatin having NORs on terminal ends. They suggest that origin of that sSMC is most likely from the heterochromatic region of chromosome no. 27. In addition, they present new information on the location and number of NORs in the Asian and savanna elephants.

It is not necessary, but I suggest that the authors could discuss the possibility that been heterochromatic and without recorded phenotypic effects, these supernumerary chromosomes can be categorized as B chromosomes.  There are more than 85 mammalian species with B chromosomes and this will be the first record of them in an elephant species.

Author Response

Reviewer 2

Thank you for reviewing the manuscript. We addressed all your comments and made changes to the manuscript.

R: It is not necessary, but I suggest that the authors could discuss the possibility that been heterochromatic and without recorded phenotypic effects, these SMC can be categorized as B chromosome.

A: We have included a paragraph in the discussion with considerations that the sSMC identified in elephants may develop into a B chromosome in the future.
